# Modelling Weather Precipitation Intensity on Surfaces in Motion with Application to Autonomous Vehicles

**DOI:** 10.3390/s23198034

**Published:** 2023-09-22

**Authors:** Mateus Carvalho, Horia Hangan

**Affiliations:** Department of Mechanical Engineering, Ontario Tech University, Oshawa, ON L1G 0C5, Canada; horia.hangan@ontariotechu.ca

**Keywords:** autonomous vehicles, weather, automotive, modeling, precipitation, rain, snow, dimensional analysis

## Abstract

With advances in the development of autonomous vehicles (AVs), more attention has been paid to the effects caused by adverse weather conditions. It is well known that the performance of self-driving vehicles is reduced when they are exposed to stressors that impair visibility or cause water or snow accumulation on sensor surfaces. This paper proposes a model to quantify weather precipitation, such as rain and snow, perceived by moving vehicles based on outdoor data. The modeling covers a wide range of parameters, such as varying the wind direction and realistic particle size distributions. The model allows the calculation of precipitation intensity on inclined surfaces of different orientations and on a circular driving path. The modeling results were partially validated against direct measurements carried out using a test vehicle. The model outputs showed a strong correlation with the experimental data for both rain and snow. Mitigation strategies for heavy precipitation on vehicles can be developed, and correlations between precipitation rate and accumulation level can be traced using the presented analytical model. A dimensional analysis of the problem highlighted the critical parameters that can help the design of future experiments. The obtained results highlight the importance of the angle of the sensing surface for the perceived precipitation level. The proposed model was used to analyze optimal orientations for minimization of the precipitation flux, which can help to determine the positioning of sensors on the surface of autonomous vehicles.

## 1. Introduction

The Society of Automotive Engineers (SAE) defines different levels of automation for road vehicles, ranging from absence of control systems to self-driving capability. While advancements in the field of autonomous vehicles (AVs) are at a developed stage, weather factors make complete autonomy in realistic situations unfeasible. AV perception systems are based on a class of embedded sensors called advanced driver assistance system (ADAS). This group of devices includes technologies such as RADAR, LiDAR, ultrasonic sensors, cameras, and global navigation satellite systems (GNSS) [1]. Recognition of the surroundings by an ADAS is compromised once the vehicle is exposed to conditions such as precipitation, fog, and lightning. For this reason, car manufacturers advise against full vehicle autonomy without driver supervision under adverse weather conditions. Concerns about road safety have increased with the expected growth in the number of AVs with the maximum level of autonomy available on the market [2]. Reports have been produced in an effort to monitor the cause of road accidents involving AVs [3,4].

In addition to driver safety and comfort, the interest behind the development of efficient ADAS is the creation of intelligent road management systems. For this, new technologies have been studied to improve the performance of sensors in conditions such as those encountered on roads. Van Brummelen et al. [5] provided a review of perception systems for AVs, while Vargas et al. [1] focused on their vulnerability to weather stressors. In the case of rain, when the diameter of the droplets is similar to or greater than the transmission wavelength, Mie scattering can be observed [6]. LiDAR sensors are particularly affected by this effect. As a result, the drops can cause attenuation of the electromagnetic signal emitted by recognition sensors, since the particles can absorb energy from the waves passing through them. Another possible impact is signal distortion, which can lead to false alarms. When it comes to snow, accumulations on sensor surfaces compromise ADAS performance. This issue has been extensively studied in the field of aviation, where autonomous systems have been in operation for longer. It is well known that the formation of ice on pitot tubes has been the cause of several plane crashes [7]. As the automotive industry moves towards self-driving solutions, the concern about icing on vehicles increases.

Recent studies have focused on the operation of AVs under precipitation conditions. Regarding mitigation methods for adverse weather, Hnewa and Radha [8] described emerging object recognition algorithms under rainy conditions using CNNs (convolutional neural networks). As for data collection and road management, Dey et al. [9] addressed the potential of intelligent transport systems (ITS), such as vehicle-based sensors and the creation of public databases. To use moving cars as rain gauges, Rabiei et al. [10] designed a system that uses the windshield wipers to estimate rainfall intensity. Drobot et al. [11] adapted commercial vehicles for monitoring road conditions during and after episodes of heavy precipitation. Moreover, innovative compact weather stations have been developed using unmanned aerial vehicles (UAVs). Among the wide range of applications that these provide, studies of the atmospheric boundary layer profile and air quality monitoring can be mentioned [12]. Road management is still a minority subject in studies involving UAVs, but low cost multicopters have been used to collect wind information using acceleration data [13,14]. This concept has gained strength with the development of climate-measurement-oriented drones by Meteomatics [15]. It is therefore possible to imagine intelligent systems for mitigation of weather impacts on AVs that combine data collected in real time by embedded sensors and weather information recorded by static referents, such as compact meteorological stations or UAVs.

The WoW (Weather on Wheels) project is part of a Canada Research Chair program in adaptive aerodynamics at Ontario Tech University and the Automotive Centre of Excellence (ACE), to seek solutions to the problems previously mentioned [16]. To accomplish this objective, the project follows a framework composed of three steps. First, full-scale measurements and numerical (CFD) simulations are conducted to derive statistically significant critical weather conditions. A method involving principal component analysis (PCA) and unsupervised machine learning has been developed to classify weather data obtained with the test vehicle of the project [17]. Next, weather conditions considered as strategic are set to be simulated in the ACE climatic wind tunnel at Ontario Tech University. Pao et al. (2023) [18] provide a complete overview of wind tunnel testing methodologies for analyzing weather effects on optical sensors of AVs. The present paper proposes a theoretical approach to the problem of perceived precipitation by moving vehicles and is included in the first phase of the project. The progress made represents an important step forward in the development of weather impact prediction models for AVs based on data collected experimentally.

The development of effective weather mitigation systems depends on the analysis of climate conditions from the vehicle’s perspective. The action to be taken by the AV must rely on the optimization of a metric recorded by the on-board system. For this, a model to quantify the perceived precipitation rate during motion must be built. In this paper, the issue of rainfall perceived by a moving vehicle is investigated, to determine a parameter that can be used as a cost function to be minimized. Real weather conditions measured in southern Ontario were incorporated into simulations to calculate the precipitation rate over surfaces with various orientations. A data-driven method with unsupervised machine learning was used to select the strategic parameters [17]. In addition, the critical variables of the problem were identified through dimensional analysis, to explain the physics of the phenomenon and help prepare for future field measurement campaigns.

## 2. Precipitation Flux Model

To mitigate the effects of heavy precipitation on the operation of autonomous vehicle sensors, an action strategy must be defined based on a metric recorded in real time. Considering only sensing surfaces, an analogy between this problem and the issue of running or walking in the rain, so as not to get wet, can be made. The literature provides different methods for establishing the optimal speed to reduce the volume of water encountered by moving objects in linear trajectories. Stern [19], for example, stated that it is necessary to minimize the number of drop hits along the body’s path. He presented a method that calculates the number of drops swept by projections of a moving surface with a given orientation. The purpose of the theoretical work was not to establish a metric for quantifying precipitation levels, as no attention was devoted to experimental apparatus or data collection. Despite coming to the conclusion that higher velocities do not always reduce the intensity of precipitation on a given surface, the model did not take into account important factors such as the diameter of the particles and the respective terminal velocities. For this reason, minimizing the number of strikes is not considered the most suitable approach for applications in autonomous vehicles. De Angelis [20] can also be cited as following the thread of reducing the number of drop impacts. However, the use of oversimplified assumptions led to the mistaken conclusion that it is always better to run than to walk in the rain to not to get wet. This was due to the fact that the cases analyzed only considered movement under vertical rain or upwind, which is not representative of events observed in real life. Holden et al. [21], in turn, introduced the concept of rain flux. He proposed the minimization of the total mass of water encountered by an object in motion, but he assumed only vertical rainfall for simplification purposes. Bailey [22] calculated the total volume of water in contact with the surface of a body using a similar method. He stated that the amount of water only depends on its specific mass, the relative speed of the droplets, the surface area, and the exposure time. He defined the best strategy for different cases, according to the direction of motion and wind. Ehrmann and Blachowicz [23] described a derivation of this method for generalized application. By approximating the shape of the human body as a cylinder, they presented a model that can find the optimal speed for different test parameters. Finally, Bocci [24] formulated the problem using equations from electromagnetism, to calculate the rain flux on a moving surface. This methodology provided an approach that covered a wide range of cases. However, some adaptations need to be made, since the problem addressed in the present paper deals with particle flux. In addition, experimental methods for recording rain data were not covered. Table 1 summarizes the major aspects of the existing models mentioned.

For educational purposes, the perceived rain methods mentioned so far considered only pre-established trajectories for the moving objects, which is not always the case in real applications. A constant precipitation rate was also assumed. Moreover, since the goal is to minimize the amount of water on the body in motion, some of them stated that the best strategy is to move as fast as possible. The reason for this is that the exposure time is reduced. Increasing the speed means reducing the time in the rain. Addressing the case of AVs, this approach is not suitable, due to external factors that influence travel time and the fact that the intensity of rain or snow varies over time. To mitigate the effects of intense precipitation on vehicles in real time, another parameter for the integration of the water mass on the surface should be considered.

### 2.1. Model Description

The method described in this paper for evaluating the precipitation perceived by the test vehicle is based on the approach of Bocci [24]. However, instead of reducing the total amount of water encountered by the car, the method defines the instantaneous rain flux Φ, in kg/s, as the cost function to be minimized. This flux can be expressed as the following integral over a closed surface *A*:(1)Φ=−∮j·dA
where j is the particle density vector defined as
(2)$$j=\rho V\left(D\right)N_{d}v$$
where ρ is the specific mass of water, in kg/m^3^ and v is the resulting velocity vector between the droplets and the surface. The particle concentration number $N_{d}$, in m${}^{-3}$, and the droplets volume V(D) in m${}^{3}$, which is a function of particle diameter *D*, can be obtained experimentally through size distributions. Thus, since there is no field source and the flux over the surface *S* of a sensor is considered, Equation (Equation 1) defined on the interval [0,∞) can be written as
(3)$$\Phi =-\int_{S}j\cdot \;dS$$

The flux equation shows that the amount of rainfall crossing the sensor surface is equivalent to the projection of the rain density over the normal surface vector. This means that only the component of j that falls perpendicularly toward the sensor is relevant for flux calculation. Therefore, crosswinds are not considered in the equation, since the inner product between the *y* direction and the sensor surface vector is zero. By considering only its front surface, the problem of a vehicle moving in the rain can be represented in two dimensions, as shown in Figure 1.

The velocity of the vehicle and the droplet are represented by $v_{c}$ and $v_{d}$, respectively. The latter can be decomposed as the sum of the terminal velocity $v_{t}$ and the wind speed $v_{wind}$. Given all the problem parameters, the equation for rain flux over the rectangular, flat sensor surface can be written as
(4)$$\Phi =-\rho N_{d}V\left(D\right)v\cdot S$$
where v is the resulting velocity magnitude normal to the sensor plane:(5)$$v=(v_{c}-v_{wind_{x}},v_{t})$$

The definition of a metric through an analytical model for quantifying precipitation levels on moving surfaces allows the development of prediction models for road management and monitoring systems. A network of weather stations can be envisioned to set safety thresholds for the operation of autonomous vehicles exposed to heavy precipitation events. When dealing with snow, Equation (Equation 4) serves as the foundation for developing accretion models for the analyzed surfaces. Extensive work has been carried out to model the accumulation of snow on power lines, given that the ice loads generated during storms can lead to the collapse of energy towers. Davalos et al. [25] used neural networks and data from weather stations in British Columbia to predict accumulation levels on power lines in nearby locations that are difficult to access. This methodology can be applied to the context of autonomous vehicles on the road. In order to relate precipitation intensity to snow accretion, it is necessary to model the interaction between the particles and the surface. This is achieved by computing coefficients known as collision, sticking, and accumulation efficiencies [26]. This is not an easy task, since the value of the coefficients depends on a variety of factors, including the type of snow encountered. Despite the complexity, a series of models have been developed over the years focusing exclusively on power cables. Makkonen [27] provided an overview of the existing models. The same principles can be used for other applications, as seen in research focused on predicting snow loads on roofs [28]. After adding the appropriate efficiency coefficients, the models developed for this purpose are similar to Equation (Equation 4), except that the relative velocity vector is only equivalent to the velocity of the particles, since the surfaces are static. The development of snow accretion models based on the model proposed in the present work is foreseen in subsequent stages of the WoW project.

As the goal of this paper is to quantify the rain flux over moving objects under realistic conditions, a generalist approach is needed. Owing to this, the simulations carried out in this work were designed to cover cases dealing with various surface orientations, nonlinear trajectories, and changing wind direction. Therefore, for a broad understanding of the problem, dimensional analysis was applied to identify the critical parameters to be investigated.

### 2.2. Dimensional Analysis

The Buckingham Π theorem defines how equations describing physical phenomena can be written as relations between dimensionless groups comprising the physical parameters involved. This method has been widely used in experimental studies over the years, especially in the field of fluid mechanics, as it provides an analytical approach to reducing the dimensionality of functions describing physical systems [29,30]. To that end, thanks to increases in computational power, the Buckingham theorem has been integrated into data-driven methods in recent studies [31].

When it comes to the subject of rain flux on a moving object, the formulation of the problem made previously is important for the identification of the variables. Therefore, by analyzing Equation (Equation 4) for a surface with a given orientation θ, one can conclude that it is a function of the following variables:(6)$$\Phi =f(v,\rho ,N_{d},S,D)$$

Through dimensional analysis, *f* can be written in the form of *G*, a new function involving the dimensionless numbers:(7)$$\Pi_{1}=\frac{\Phi }{\rho D^{2}\left||v\right||}$$
(8)$$\Pi_{2}=\frac{S}{D^{2}}$$
(9)$$\Pi_{3}=D^{3}N_{d}$$
(10)$$\frac{\Phi }{\rho D^{2}\left||v\right||}=G\left(\frac{S}{D^{2}},D^{3}N_{d}\right)$$

The composition of the groups has a physical interpretation, as $\Pi_{1}$ is related to the amount of rain mass encountered, $\Pi_{2}$ is associated with the exposed surface area, and $\Pi_{3}$ represents the climatic condition in the surroundings of the object. This type of analysis shows that, to evaluate the precipitation perceived by the vehicle, it is necessary to access the mean particle size, as well as their average velocity, in addition to collecting wind data such as the speed and direction. This is important for the definition of the sensors to be employed in the measurement campaigns.

In addition, one important conclusion is that, contrary to what many articles claim, the speed of the object does not define the flux of rain over it. The resulting velocity between the body and the drops is the crucial factor, since assuming an object in motion is analogous to imagining it is a static referential exposed to wind passing over it. Given that the dimensions of the sensor surface *S* are known and its orientation depends on the object’s geometry, the critical parameters for the analysis of the problem, which were considered in the simulations, are the surrounding weather conditions, expressed by the mean particle diameter *D* and the concentration number $N_{d}$. The specific mass of water ρ can be estimated. The parameter to be simulated, therefore, is the resulting velocity between the surface and the precipitation v. Implicitly this vector contains information about the speed of the vehicle, its trajectory, and the angle that is formed between the surface vector and the wind in each time step.

## 3. Model Validation

To validate the proposed rain flux model, outputs from Equation (Equation 4) fed with experimental data were compared with direct rainfall rate readings. Data were recorded using a road vehicle adapted with sensors able to measure weather information, such as temperature, relative humidity, and particle size, as well as velocity distributions. The test vehicle was developed by ACE’s technical team and measurements were carried out at the Canadian Technical Center McLaughlin Advanced Technology Track (CTC MATT) located at GM’s assembly plant at Oshawa, Ontario. In this section, the experimental apparatus used in the measurement campaigns is described, along with the processing of the sensors’ data. Then, a comparison between the results obtained with the theoretical model and the direct measurements coming from the sensors is presented.

### 3.1. Experimental Setup

The vehicle was equipped with a Vaisala FD70 optical disdrometer that provided 60 s histograms of particle size and velocity. Wind data were sampled at 5 Hz using a Campbell CSAT3 3D sonic anemometer. Data were recorded in three orthogonal directions, which were adjusted to match the coordinate system of the simulations. A Vaisala WXT530 weather transmitter was used to measure temperature and relative humidity every 5 s. The equipment was installed on a rack above the car specially built for the measurements, as shown in Figure 2. More details about the experimental setup were provided by Carvalho and Hangan [17].

### 3.2. Data Processing

Optical disdrometers allow the quantification of the level of precipitation in different ways. Frasson et al. [32], for example, estimated rainfall rate from outdoor measurements by summing the volume of drops. Based on this approach, the total volume of precipitation found in the experimental samples was calculated as
(11)$$V\left(D\right)=\sum_{j=1}^{t}\sum_{i=1}^{k}N_{i,j}\frac{4\pi }{3}{\left(\frac{D_{b_{i,j}}}{2}\right)}^{3}$$
where *N* is the count number of each size bin of mean diameter $D_{b}$. As the equation shows, it is common practice in disdrometer data processing to approximate the droplet volume by considering them as spheres. Parameters such as *k*, the total number of size bins, depend on the disdrometer model. The count number *N* is used for the calculation of the concentration number $N_{d}$ [33] for each size distribution.
(12)$$N_{d}=\frac{1}{||v_{n}||S\Delta t}\sum_{i=1}^{k}N_{i}$$

The magnitude of the droplet net velocity $v_{n}$ can be obtained from the weighted average of the droplet speed distributions.
(13)$$\left||v_{n}\right||=\frac{\sum_{i=1}^{k}N_{i}v_{d_{i}}}{\sum_{i=1}^{k}N_{i}}$$

### 3.3. Validation Results

Data collected under snow and rain, respectively, on 7 and 23 March 2022 on the GM (General Motors) Canada McLaughlin Advanced Technology Track (MATT) were used for validation. This specific dataset relied on 1 s droplet size distribution (DSD) readings from the FD70 disdrometer with its sensing surface in a horizontal position. The measurements were made over a period of 8 min, and three cases were considered regarding the driving speed. First, the car was stationary during data collection, similar to a standard meteorological tower. Then, measurements were carried out while the test vehicle traveled along the track at 40 km/h and 80 km/h.

The disdrometer provided direct precipitation intensity outputs in mm/h. Being an optical sensor, the FD70 processed the drop size distributions internally, in order to generate these signals. The model validation procedure therefore sought to verify if the rain flux equation obtained signals that were comparable to those measured by the sensor. The results are shown in Figure 3 for rain data and in Figure 4 for snowfall. For all tested cases, the time signals showed a strong correlation, indicating that the model performed well for both stationary and moving vehicles. This showed an agreement with the prediction using flux theory, where the only component that mattered is the normal to the analyzed surface. For the case of the horizontal sensing plane, this component was the droplet drop velocity, which does not depend on the driving speed.

To quantify the performance of the model in comparison with the sensor measurements, a correlation study between the signals was performed. Table 2 shows the values of the Pearson coefficients [34]. As was seen in the time series, the signals showed a strong correlation, especially in the case of rainfall, where the coefficients were always greater than 0.90. As expected, the model had a worse performance when it came to snow. Snow particles are more susceptible to aerodynamic forces, and have a varied morphology that depends on the surrounding weather conditions [35]. While these aspects were not taken into consideration in the present study, it is important to investigate in detail their impact, to make the model more accurate.

Despite the similarity in the trends, a calibration experiment of the disdrometer should be done to compare the precipitation mass values. The analysis of the validation results was limited to the computation of the correlation coefficient, since the processing chain developed by the manufacturer was unknown. In the case of snow precipitation, this process is more complex, since mass and density estimation models as a function of particle diameter have to be considered to determine the accumulation levels [36].

Although the correlation between theoretical and experimental data was strong, the error observed in some cases was not negligible. This can be attributed to resolution issues affecting the sensor, due to the low intensity of precipitation encountered during the experiments. To address this, a larger dataset should be used in the following stages of the project, to evaluate the model’s performance over a wide range of precipitation levels. Nevertheless, the results are considered satisfactory for simulation purposes, while future experimental campaigns are being prepared.

## 4. Evaluation of Perceived Precipitation

The comparison between the magnitude of the values obtained by the model and by the sensor validated the rain flux calculation method through Equation (Equation 4). Scenarios that were more complex could then be tested. Simulations were performed considering a body moving along a circular path with a perimeter of 3.6 km, the same as the test track where the experiments were carried out. The rain flux was calculated on three surfaces with different orientations. Parameters such as the concentration number $N_{d}$ and total droplet volume *V* were estimated from the field measurement campaign mentioned previously, using a machine learning-based method described by Carvalho and Hangan [17]. Wind speed $v_{wind}$ in three directions came from anemometer measurements. The sensing surface had the same dimensions as the FD70 disdrometer.

### 4.1. Simulation Parameters

The works cited in the introduction to Section 2 simulated different cases of objects moving along a linear path under rain. Since the goal of this paper was to define optimal traveling speeds, the difference between the simulation cases basically consisted of the wind direction. The previously studied configurations were limited to situations where the object was moving towards or in the same direction as the precipitation, as well as vertical rain. Here, to cover a wide range of cases in the same simulation, the resulting vector v was modeled as a function of the components $v_{c}$, the vehicle speed, and $v_{d}$, the droplets velocity vector. The object trajectory was designed to be circular, in the xy plane. This choice was made as a compromise between the real trajectory of the road vehicle, which was an oval test circuit, and the simplicity of the analytical equation of a circle.

Spherical coordinates x=(r,ϕ,θ) were used to model the rain flux along the object’s motion. First, the angular position ϕ(t) was defined as being linearly varied over time:(14)$$\phi \left(t\right)=\phi_{0}+\frac{\;d\phi }{\;dt}t$$

The object displacement vector r(t), which describes the position of the vehicle as a function of time in terms of angle ϕ is:(15)$$r\left(t\right)=r\cos\left(\phi \right)\hat{i}+r\sin\left(\phi \right)\hat{j}$$

Consequently, the velocity vector $v_{c}$, which was the direction of the vehicle’s displacement along the trajectory, was obtained using a derivative of Equation (Equation 15):(16)$$v_{c}\left(t\right)=r\frac{\;d\phi }{\;dt}\left[-\sin\left(\phi \right)\hat{i}+\cos\left(\phi \right)\hat{j}\right]$$

The particle velocity $v_{d}$ is the sum of the wind speed components and the droplet terminal velocity. Regarding the droplets terminal velocity $v_{t}$, the model obtained experimentally by Atlas et al. [37] for vertical precipitation was adopted. The three-dimension vector is therefore:(17)$$v_{t}\left(D\right)=\left(0,0,-9.65+10.3e^{-0.6D}\right)$$

Since the precipitation conditions used in the simulation did not vary, the terminal velocity of the droplets was also constant. To create a time-varying aspect for the terminal velocity, the assumption that the vertical component of the wind was embedded in the value generated by the model was made. With $v_{d}$ being the sum of the wind vector $v_{wind}$ and the droplet terminal velocity $v_{c}$, the rain flux Φ could be found by projecting the resultant vector formed with the vehicle vector $v_{c}$ onto the direction normal to the sensing surface, as shown in Equation (Equation 4).

### 4.2. Setting Realistic Parameters for Simulation

The proposed theoretical rain flux model opens the door for various applications concerning AV safety. Numerical simulations that allow the quantification of perceived precipitation on moving objects can help the development of mitigation solutions for the operation of ADAS sensors. In addition, reinforcement learning (RL) techniques could be applied to train an agent to minimize the impact of weather stressors by considering the rain flux equation as a cost function.

To generate realistic results, simulations were performed based on weather conditions measured by the test vehicle at GM’s MATT facilities on 23 March 2022. Since the goal was to compute the rain flux on surfaces with different orientations, it was decided to use real wind data from the CSAT3 sonic anemometer in the simulations. Thus, the wind was expected to play a determining role in the flux value, unlike the validation tests, where the analyzed surface was oriented horizontally. The time series of the vector $v_{wind}$ components can be seen in Figure 5. The anemometer data were recorded with the car stopped over a period of 10 min. Wind data can also be generated synthetically using periodic functions, as described in Appendix A. By combining real wind data with the terminal velocity model from Equation (Equation 17), the velocity vector of the drops could be generated.

The next step was to implement the concentration number $N_{d}$ in the simulation loop and to set the total droplet volume V(D). To match the duration of the wind signals, it was assumed that the precipitation conditions remained constant throughout the test. To do so, realistic values of $N_{d}$ and V(D) were computed from the measured droplet size distributions from the experimental campaign. A data-driven method was employed to establish strategic parameters for the simulation [17]. The process consisted of fitting normalized DSDs with a theoretical model that combined the exponential distribution of Marshall and Palmer [38] with a Gaussian therm. Then, a data matrix containing the parameters of the fitted functions was created and principal component analysis (PCA) was employed to reduce the dimensionality. Finally, a K-means model was used to define clusters that contained elements with similar features. Statistical analysis of the formed clusters allowed the identification of average parameters that represented recurrent local types of precipitation, including drizzle, freezing rain, and snowfall.

This method was applied to the entire rainfall dataset formed throughout the project duration. Figure 6 shows the distribution of variance among the principal components of the data. The PCA showed that more than 80% of the total variance was retained when considering the first three modes. To define the number of clusters for the K-means model, the elbow method was used. The curve in the Figure 7a shows that the variance within clusters started to converge when the number of groups was approximately six. Figure 7b shows the 2D scatter of the datapoints with their respective cluster.

Since it contained the most samples, corresponding to 28.85% of the total points, cluster 0 was chosen as the reference for setting the simulation parameters. By analyzing the parameter values of the theoretical model among its samples, it was possible to reconstruct a DSD that summarized the characteristics of the cluster. It is important to mention that the generated distribution was normalized, so it was necessary to use an experimental sample to scale it. The criterion for the selection of this sample was the precipitation rate at the time of measurement. The sample of cluster 0 collected with the highest precipitation intensity value was chosen. Figure 8 shows a boxplot of the DSD model parameters of cluster 0 and the result of the reconstructed distribution. The operations described in Section 3.2 were applied for the processing of the synthetic DSD to compute parameters such as the concentration number $N_{d}$ and the total droplet volume V(D).

## 5. Simulation Results

The described weather parameters were set to the same time scale and the simulations were launched for the circular trajectory mentioned at the beginning of the section in a counterclockwise direction. The rain flux on three surfaces was computed for each time step of the simulation loop, in order to generate individual time series for each orientation. The values of θ defined by the spherical system of coordinates used in the calculations were 90°, 45°, and 0°.

The simulation results for driving speeds of 40 and 80 km/h, which were the same as those used in the data collection on the MATT track, are shown in Figure 9. The flux values are compared for surfaces with the same orientation, but at different driving speeds (Figure 9a), and test cases where the driving speeds are the same, but the angles vary (Figure 9b). The first noticeable aspect of the time series is their periodicity as the surfaces alternate moments, where they are directly exposed to wind and when they are shielded from it. This is due to the projection of the resultant vector *v* onto the vector normal to the surface *S*. When the vehicle traveled upwind, the dot product between the vectors reached a higher absolute value, while traveling downwind generated a lower flux. Consequently, the bell-shaped curves are wider at 40 km/h, as it took longer to complete a full lap. On the other hand, the magnitude of the flux was larger when the driving speed was higher, which can be explained by the fact that the object swept more particles along the path. This can also be seen in Figure 9b, where the flow at 90° becams higher compared to the 45° angle when the surfaces were moving at 80 km/h. The velocity vector of the vehicle determined the direction in which the precipitation projection reached its maximum value.

For the horizontal surface, the flux values were the same for the two tested cases, since only the drop velocity component normal to the surface was relevant to the model. In real life, the aerodynamics of the vehicle would influence the flux dynamics. Therefore, to make the model more accurate, data-driven methods are being developed to reduce the error arising from air turbulence around the vehicle surface. Experimental campaigns are being prepared using meteorological towers as a static referential in combination with the test vehicle to predict the level of precipitation experienced by the latter along complex trajectories.

The fluxes over surfaces with different orientations behaved in complex ways, as they depend on variables that are susceptible to abrupt changes over time. Therefore, other driving speeds were tested under the same weather conditions, and the flux over the surfaces where θ=45° and 90° was integrated over time. The goal was to identify possible optimal driving speeds for minimizing the mass of encountered precipitation. Figure 10 shows the results of the simulation for velocities ranging from 20 to 120 km/h. The accumulated rain mass shows that, when an exposure duration is imposed, the task of finding the optimal speed is not straightforward. As an example, driving at 40 km/h proved to be more efficient than driving at 20 km/h for a duration of between 350 and 450 s. In addition, the optimal orientation angle also depends on the driving speed. While the 90° angle showed lower flux values at low velocities, the same was not true for driving speeds above 40 km/h. In the case at 120 km/h, the rain mass over the vertical surface was 20% higher than over the plane at 45°. This issue is pertinent for the placement of sensors on AVs, especially when they are meant to operate at high speed, such as on highways.

The results presented so far indicate that the optimal speed for traveling through rain must be evaluated instantaneously according to the weather conditions faced. The resultant vector between the vehicle displacement and the rain velocity dictates the value of the flux. In view of the complexity of the problem, Figure 11 aims to synthesize the various simulated scenarios, showing the maximum values of precipitation mass encountered for the different surface orientations and driving speeds after 600 s. As can be seen, for all cases tested, the model generated continuous curves that intersect at θ=0° and 180°. As discussed earlier, since the model does not account for the aerodynamics of the object, the flux value over horizontal surfaces does not change as a function of the driving speed. The curves also show that the flux is always lower over the rear. As driving speed increases, flux decreases over surfaces with a more negative slope. The maximum points do not occur at the same angle, and they tend to approach 90° as the velocity increases. Finally, surfaces with normal vectors that point to the ground (θ>90°) can offer effective solutions for minimizing the level of precipitation over surface-mounted sensors, as they show reduced values of total rain mass that tend to zero as the angle increases.

## 6. Conclusions

In this paper, a model to quantify the level of weather precipitation perceived by moving vehicles was proposed. An analytical equation that predicts precipitation intensity over moving surfaces under rain or snow can serve as the foundation for data-driven methods to mitigate the negative effects of weather stressors on the operation of autonomous vehicles. The model could provide a physical background to prediction models for road safety management purposes. Moreover, the equation could serve as a cost function to be minimized by an agent exposed to weather precipitation conditions. Possible actions such as the adjustment of the traveling speed, trajectory, and the location of sensors on strategic surfaces could be considered.

Up to this point, the issue of moving in the rain has been addressed to set optimal traveling speeds for linear paths under a constant precipitation rate. Despite being consistent in their proposals, the methodologies created so far use oversimplified hypotheses that cannot be considered in AV-oriented applications. Differently from previous studies, the present simulations targeted more realistic parameters regarding weather conditions and non-linear trajectories. In addition, dimensional analysis was used to identify critical parameters for the problem. The non-dimensional groups found assisted in the design of the simulations, as well as future experiments.

The rain flux model was based on electromagnetism equations but adapted to the issue of particle flow. It was conceived in such a way as to integrate data that can be obtained experimentally. The experimental apparatus used for the measurements, as well as the processing chain of the sensor data, is described in the paper. Partial validation tests were performed comparing direct measurements of precipitation rate with values coming from the model. The results presented a strong correlation between theoretical and experimental values, showing that the model performed well for both rain and snow scenarios. Calibration experiments for the optical sensors that will be used in future campaigns were considered. Due to the orientation of the sensor during the experimental campaign, a comparison could only be made for the case where the surface was horizontal. Simulations based on real wind data highlighted the dependency of rain flux on wind direction, while the integration of the results showed that the definition of the optimal travel speed depends on the projection of the resultant vector between vehicle and particle speeds over the sensing surface. However, it is known that aerodynamic forces can influence the flux value. Therefore, the model needs further development to account for this factor. In addition, with the purpose of creating a network of meteorological stations for road management, correlation studies should be carried out between the values predicted by the model and those observed by moving vehicles. Finally, the interaction between snow particles within the boundary layer of the analyzed surfaces should be modeled, in order to adapt the precipitation intensity model for predicting snow accumulations.

As a perspective for future experimental campaigns, simultaneous measurements will be conducted between a vehicle equipped with multiple disdrometers with different orientations and a meteorological tower. The goal is to analyze data from a static referential combined with data collected in real time by the car. In addition, the future tests will aim to evaluate the influence of vehicle aerodynamics on the rain flux calculation. With this, it is expected that further layers of complexity could be incorporated into the model presented in this present paper. With more accurate results, data-driven prediction methods using meteorological stations can be envisioned as part of an intelligent road management system aimed at improving the safety of drivers.

## Figures and Tables

**Figure 1 sensors-23-08034-f001:**
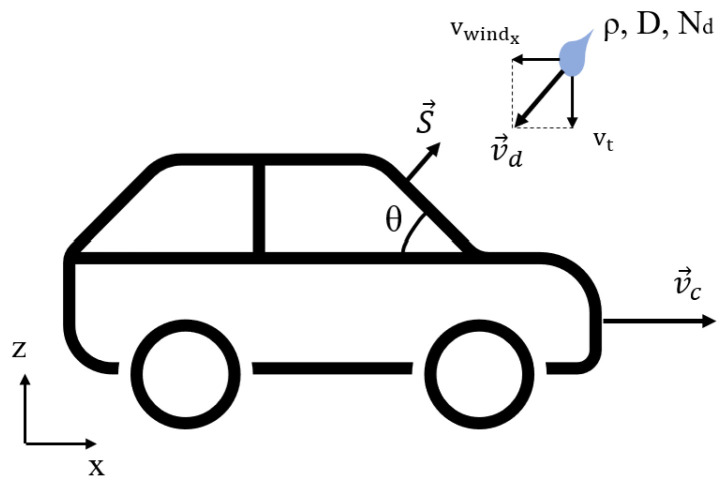
Two-dimensional representation of a moving vehicle in the rain with the sensing surface *S* oriented at θ degrees.

**Figure 2 sensors-23-08034-f002:**
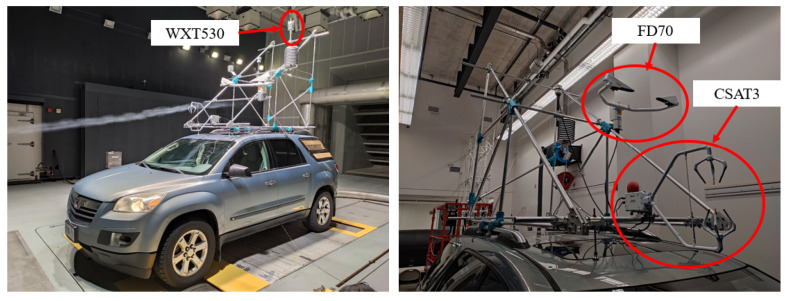
The WoW vehicle used for field measurements.

**Figure 3 sensors-23-08034-f003:**
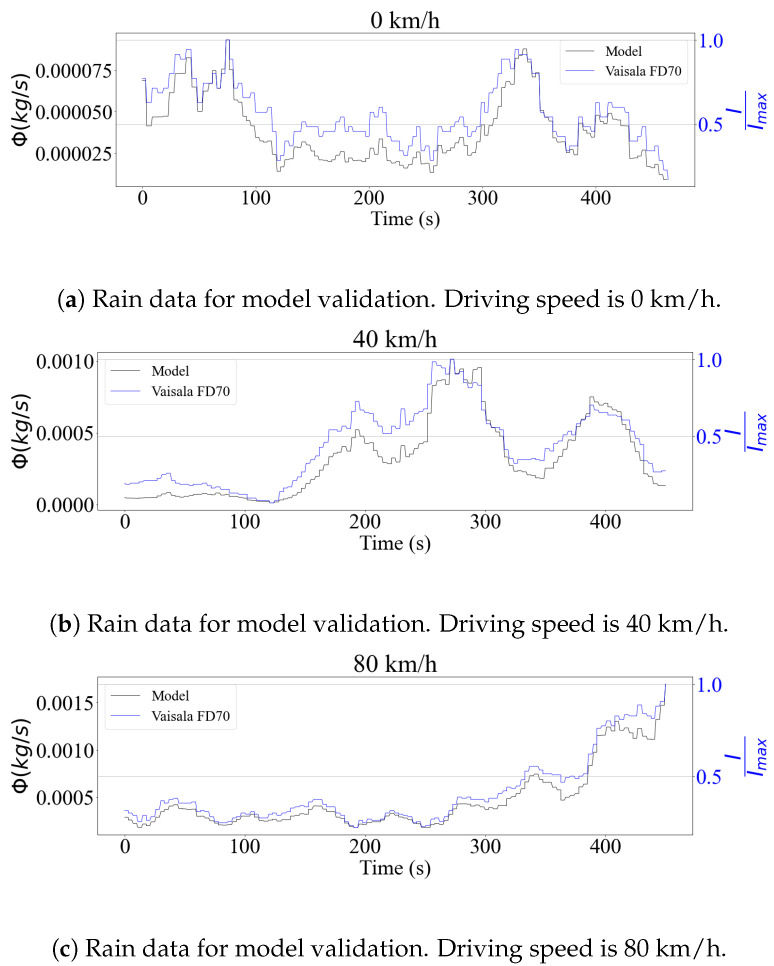
Comparison between scaled readings of rain precipitation intensity, represented as *I*, from the Vaisala FD70 disdrometer and the theoretical flux model on a horizontal surface for different driving speeds.

**Figure 4 sensors-23-08034-f004:**
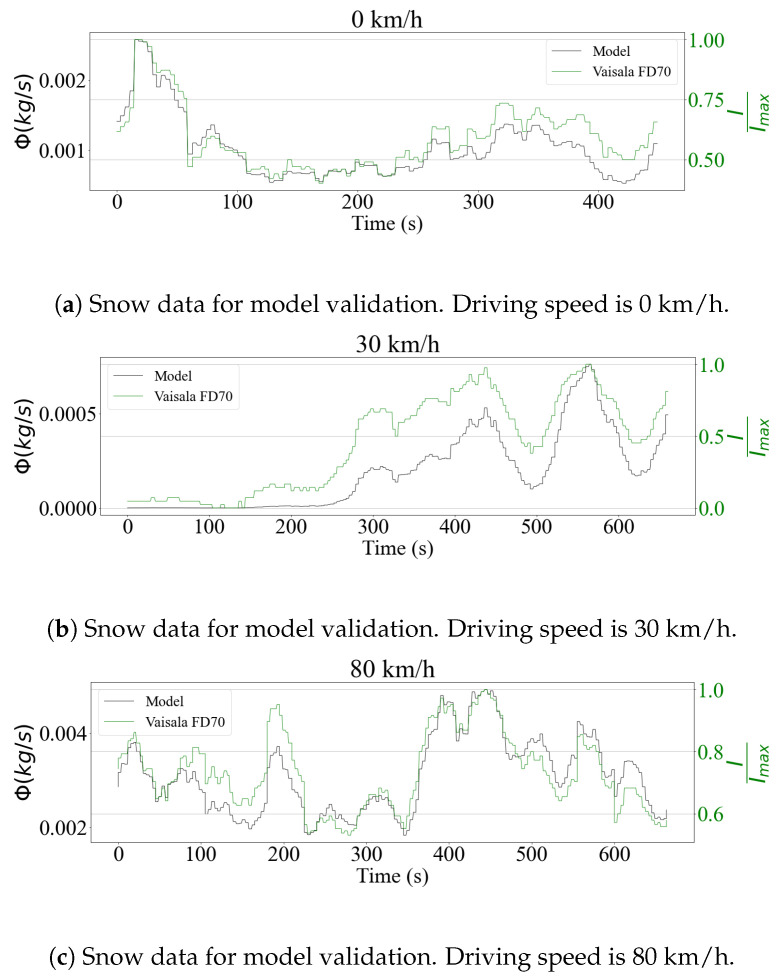
Comparison between scaled readings of snow precipitation intensity (*I*) from the Vaisala FD70 disdrometer and the theoretical flux model on a horizontal surface for different driving speeds.

**Figure 5 sensors-23-08034-f005:**
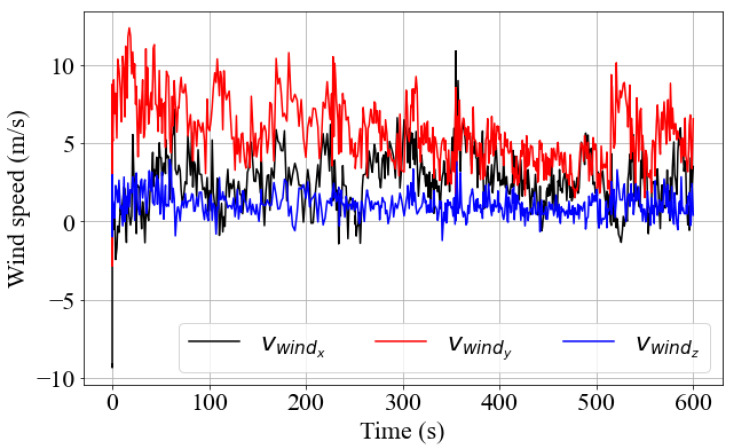
Wind speed components recorded with a CSAT3 sonic anemometer.

**Figure 6 sensors-23-08034-f006:**
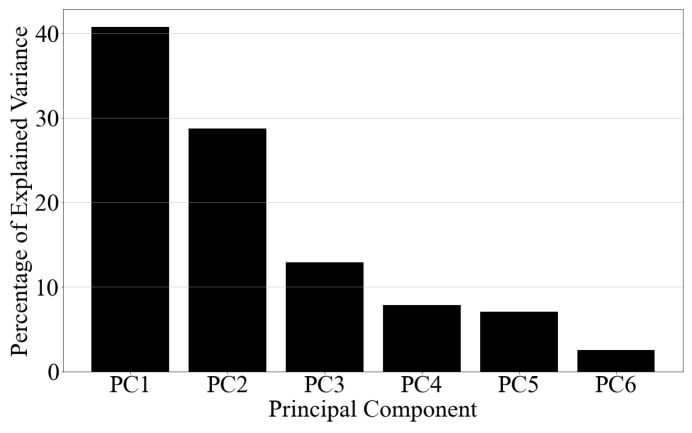
Variance of principal components. The chart shows that approximately 70% of the total variance was contained in the two first PCs.

**Figure 7 sensors-23-08034-f007:**
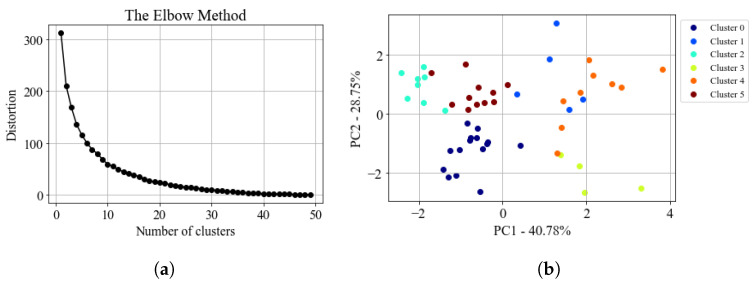
(**a**): Elbow curve used to choose the number of K-means clusters. The distortion was obtained using the sum of the squared distances between the points and their centroid within the clusters. (**b**): Two-dimensional, scatter of data points with the indication of their respective cluster.

**Figure 8 sensors-23-08034-f008:**
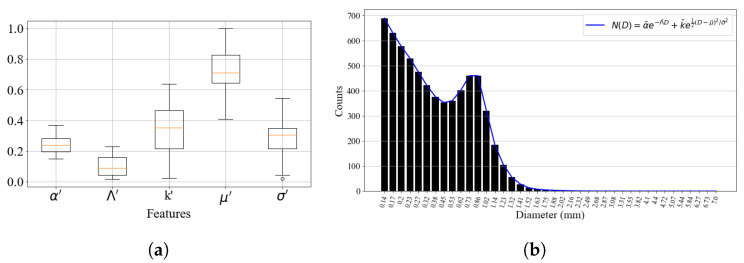
(**a**): Boxplot of the model parameters contained in cluster 0. The orange line indicates the median of the parameters. (**b**): Synthetic 60-s DSD generated from the average model parameters of cluster 0. The blue curve shows the theoretical model used to build the distribution.

**Figure 9 sensors-23-08034-f009:**
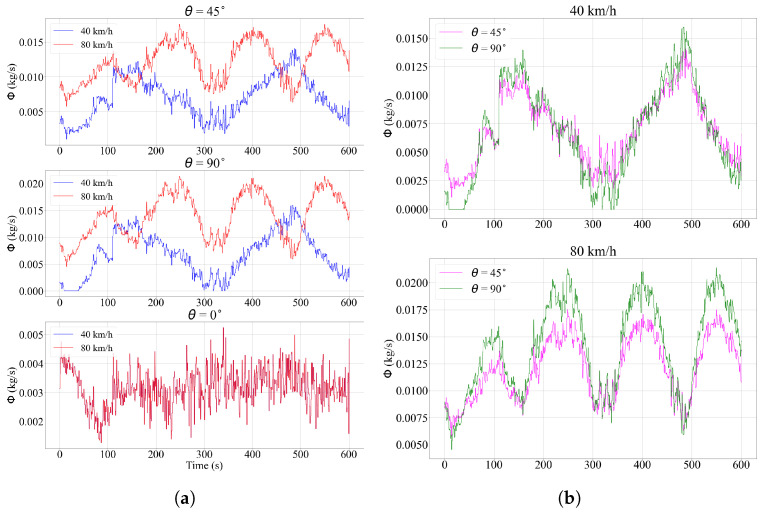
Results of simulation using real wind data. (**a**): For the same surface orientation, the rain flux was compared at driving speed, 40 and 80 km/h. (**b**): For the same driving speed, values of rain flux were compared at different angles.

**Figure 10 sensors-23-08034-f010:**
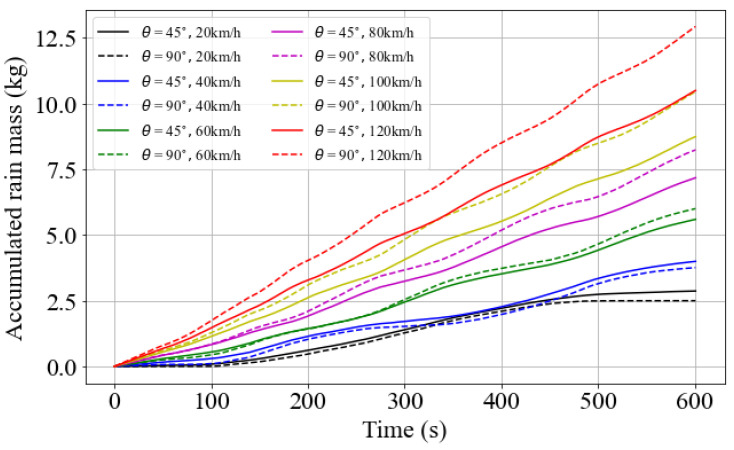
Integral of flux over time for different driving speeds and surface angles.

**Figure 11 sensors-23-08034-f011:**
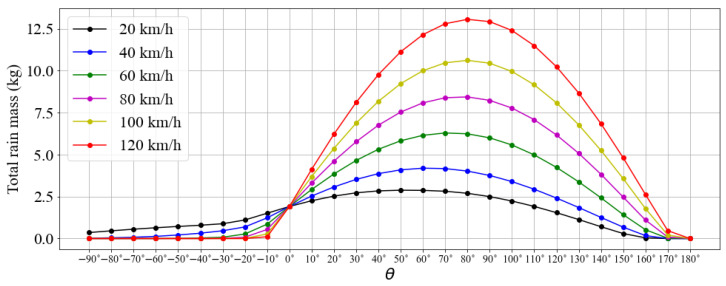
Total rain mass encountered as function of the surface orientation.

**Table 1 sensors-23-08034-t001:** Existing models for estimation of precipitation levels on moving surfaces.

Model	Approach	Comments
Stern (1983) [19]	Drop strikes	The model assumes constant rainfall conditions Rainfall volume is not calculated
de Angelis (1987) [20]	Drop strikes	The model assumes constant rainfall conditions Rainfall volume is not calculated Misconception that it is always better to move fast in the rain
Holden et al. (1995) [21]	Rain flux	The simulations only consider vertical rain Experimental aspects are not considered
Bailey (2002) [22]	Rain flux	Introduces the concept of relative droplet velocity Different expression for upwind and downwind motion
Ehrmann and Blachowicz (2011) [23]	Rain flux	Simulations assume constant rainfall conditions Experimental aspects are not considered
Bocci (2012) [24]	Rain flux	It is based on equations from electromagnetism Simulations assume constant rainfall conditions Experimental aspects are not considered
Carvalho and Hangan (2023)	Rain flux	It is based on Bocci’s model Deals with the issue of particle flux Relies on experimental data for validation

**Table 2 sensors-23-08034-t002:** Pearson correlation coefficients between the model results and FD70 measurements.

Rain			Snow		
**0 km/h**	**40 km/h**	**80 km/h**	**0 km/h**	**30 km/h**	**80 km/h**
0.95155	0.95422	0.99241	0.93424	0.91836	0.88894

## Data Availability

The data presented in this study are available on request from the corresponding author. The database is not publicly available, due to privacy issues.

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
