# Peer review of "Modelling Weather Precipitation Intensity on Surfaces in Motion with Application to Autonomous Vehicles"

_sensors, 2023, doi:10.3390/s23198034_

Round 1

Reviewer 1 Report

The authors are focused on concerns about autonomous vehicles during extreme weather conditions. The quality of the presentation and the significance of the content is good. There are a few minor changes that need to be addressed:

1) The title of the article can be improved, as the main keyword autonomous vehicles is missing in the title. Actually, the title should summarize the purpose and aim of the research.

2) A comparative analysis table must be included in the article for comparing the previous studies.

The article is well-written and it can be accepted after minor changes.

Reviewer 2 Report

The goal of the research, titled "Understanding the Effect of Rain and Snow on Vehicles in Motion," is to create a theoretical method that measures how precipitation affects moving vehicles, especially self-driving cars. The study aims to analyze actual weather conditions as experienced by the vehicles to improve a key metric recorded by their onboard systems. This model intends to help us better grasp how precipitation is perceived while a vehicle is in motion. This understanding can offer valuable information for developing strategies to deal with challenging weather and for deciding where to place sensors on self-driving cars to make sure they perform well even in tough weather.

Comments:

1.   Abstract and Introduction:

·         The study touches on how adverse weather affects autonomous vehicles (AVs), but it doesn't go into detail about how different weather conditions might impact them differently. This could leave readers wanting a better grasp of the various weather challenges that AVs encounter.

2.   WoW (Weather on Wheels) Project:

·         The project's framework is laid out in three steps, but the excerpts only cover the first one. This could create confusion about the remaining steps and the overall approach of the project.

3.   Precipitation Flux Model:

·         They use the analogy of "running or walking in the rain" to explain the model's approach. However, this might oversimplify the intricacies of AV operations. Furthermore, they mention methodologies like those by Stern and De Angelis without fully explaining them, which might leave readers unfamiliar with those references feeling a bit lost.

4.   Model Description:

·         The study cites other works that tackled the challenge of moving in the rain. They criticize some of these methods for using "oversimplifying hypotheses." While this criticism is valid, the study could benefit from a more thorough comparison, pinpointing exactly where and how these methods fall short.

5.   Data Availability:

·         The statement "Due to privacy and ethical concerns, neither the data nor the source of the data can be made available" raises worries about the study's ability to be replicated. Without access to the data, it's difficult for other researchers to externally validate the findings.

6.   General Observations:

·         Certain parts of the study reference other works without giving a detailed explanation or context. This could be confusing for readers who aren't familiar with those references. Moreover, the study could gain from a more in-depth discussion about the practical implications of the findings, especially in real-world scenarios involving AVs.

7.   The authors should add table for mathematical symbols.

8.   The authors should explain in details the proposed mathematical model.

9.   What are the limitations of proposed method? Please answer the question in the article.

10.         Please  follow the journal template.

Reviewer 3 Report

This manuscript targets a model to quantify precipitation of rain or snow for the moving vehicle and validated by outdoor data. Parameters such as wind direction, particle size were further investigated. Overall, it is a nicely organized and well-written paper, with sufficient technical details and solid validations. It is a completed research and I am glad to recommend it to be accepted in its current form to the journal. Thank you for this interesting research and good luck to your future works.

Reviewer 4 Report

1. The third page of the manuscript mentions 'Equation 1 defined on the interval [0, [can be written as: '. Please note the use of '[' and ']'.

2. There is a unit format issue on the third page of the manuscript, such as kg/m3. Please check if the author needs to adjust it to kg/m3.

3. From Table 1 of the manuscript, it can be found that the signals show strong correlation, especially in the case of rainfall, but there are still small gaps. It is recommended that the author analyze and improve the influencing factors that cause this small gap.

4. The pictures shown in the manuscript are rather blurred, such as Figure 6. It is suggested that the author should replace or adjust them.

The authors should be rechecking the grammatical errors and typos issues in the complete manuscript.

Round 2

Reviewer 2 Report

The goal of the research, titled "Understanding the Effect of Rain and Snow on Vehicles in Motion," is to create a theoretical method that measures how precipitation affects moving vehicles, especially self-driving cars. The study aims to analyze actual weather conditions as experienced by the vehicles to improve a key metric recorded by their onboard systems. This model intends to help us better grasp how precipitation is perceived while a vehicle is in motion. This understanding can offer valuable information for developing strategies to deal with challenging weather and for deciding where to place sensors on self-driving cars to make sure they perform well even in tough weather.

Comments:

The authors made the correlations perfectly just need to follow the journal template.

congratulations

Reviewer 4 Report

I thank the authors for their updates. The paper has been considerably improved. However, I still recommend a MINOR REVISION. This is because the overall quality of the English text has improved but is not perfect. Moreover, some minor issues remain in the manuscript, as follows.

1. Some cited references are too old.

2. The formula (6-10) in the manuscript contains parameter D, but the manuscript only mentions that the surrounding weather conditions are represented by D and Nd, without providing the specific meaning and data form of parameter D. It is recommended that the author supplement it.

None.
